# Microbial communities can predict the ecological condition of headwater streams

Robert H. Hilderbrand[1]*, Stephen R. Keller[1¤], Sarah M. Laperriere[2,3,4], Alyson E. Santoro[2,3], Jason Cessna[1], Regina Trott[1]

**1** Appalachian Laboratory, University of Maryland Center for Environmental Science, Frostburg, MD, United States of America, **2** Horn Point Laboratory, University of Maryland Center for Environmental Science, Cambridge, MD, United States of America, **3** Department of Ecology, Evolution, and Marine Biology, University of California, Santa Barbara, CA, United States of America, **4** Department of Biological Sciences, University of Southern California, Los Angeles, CA, United States of America

¤ Current address: Department of Plant Biology, University of Vermont, Burlington, VT, United States of America

* rhilderbrand@umces.edu

**Data Availability Statement:** Sequence data are available from NCBI under BioProject accession number PRJNA545742.

**Funding:** Funding was provided by the Maryland Sea Grant (https://www.mdsg.umd.edu/) through award NOAA(SG):NA14OAR4170090 to RHH, SRK,

## Abstract

Humanity's reliance on clean water and the ecosystem services provided makes identifying efficient and effective ways to assess the ecological condition of streams ever more important. We used high throughput sequencing of the 16S rRNA region to explore relationships between stream microbial communities, environmental attributes, and assessments of stream ecological condition. Bacteria and archaea in microbial community samples collected from the water column and from stream sediments during spring and summer were used to replicate standard assessments of ecological condition performed with benthic macroinvertebrate collections via the Benthic Index of Biotic Integrity (BIBI). Microbe-based condition assessments were generated at different levels of taxonomic resolution from phylum to OTU (Operational Taxonomic Units) in order to understand appropriate levels of taxonomic aggregation. Stream sediment microbial communities from both spring and summer were much better than the water column at replicating BIBI condition assessment results. Accuracies were as high as 100% on training data used to build the models and up to 80% on validation data used to assess predictions. Assessments using all OTUs usually had the highest accuracy on training data, but were lower on validation data due to overfitting. In contrast, assessments at the order-level had similar performance accuracy for validation data, and a reduced subset of orders also performed well, suggesting the method could be generalized to other watersheds. Subsets of the important orders responded similarly to environmental gradients compared to the entire community, where strong shifts in community structure occurred for known aquatic stressors such as pH, dissolved organic carbon, and nitrate nitrogen. The results suggest the stream microbes may be useful for assessing the ecological condition of streams and especially useful for stream restorations where many eukaryotic taxa have been eliminated due to prior degradation and are unable to recolonize.

and AES. The sponsors had no role in the study design, data collection and analysis, decision to publish, or manuscript preparation.

**Competing interests:** The authors have declared that no competing interests exist.

## Introduction

Microbes are crucial to biogeochemical processes and cycles at all spatial scales, from intra-individual to global. They are key to the maintenance of biodiversity and ecosystem function, and identifying factors influencing microbe distributions and diversity may be important for understanding ecosystems across natural and human-influenced gradients. Advances in high throughput sequencing of environmental samples have dramatically expanded our knowledge of microbial biodiversity [1] and present substantial opportunities for the environmental sciences. For example, the diversity and composition of the microbial community may be useful as direct or indirect proxies for assessing ecosystem condition and health and have recently been used to predict hydrologic function in large Arctic rivers [2]. Microbe-based applications for ecosystem monitoring and assessment are particularly exciting for classifying the condition of freshwater streams because a small, easily collected sample could augment or replace the substantial efforts required for traditional methods based on eukaryotes [e.g., 3]. Analogs also exist for the human body and other animals, where the overall status of an individual can be inferred by the composition of its microbiome as indicators of age, disease, or other stressors [4–7].

Assessing the ecological condition of freshwater streams has become increasingly important because much of humanity relies on streams to supply water for drinking and irrigation, as well as many ecosystem services. This reliance will only increase with the growing human population, which itself results in more stream degradation [8, 9]. Because water flows downhill, streams integrate activities in the upstream catchment, including the headwaters. The identity and community structure of stream biota closely reflects the level of disturbance and degradation upstream, and various eukaryotic groups such as benthic macroinvertebrates [10, 11], fish [12, 13], and diatoms [14] are currently used by many organizations for biomonitoring. While all have their utility, benthic macroinvertebrates are probably the most relied upon because they are relatively easy to collect and identify, are ubiquitous in perennial stream ecosystems, and have several order-level taxa sensitive to pollution and known to be good indicators of stream health [15, 16]. However, the data on macroinvertebrates needed to use them as bioindicators of stream health remain time consuming to collect, process, and identify.

Because many eukaryotic groups sort along natural and human-driven environmental gradients [17, 18], microbes might be used similarly as macroinvertebrates as bioindicators of stream condition. Indeed, freshwater microbes appear to be strongly aligned with pH and dissolved organic carbon (DOC) gradients [19–21], while soil microbes show functional and structural responses to pH [22], moisture [23] and organic carbon [20]. Freshwater bacterial community structure varies with differences in land cover [24, 25] and has been correlated with more traditional indicators of stream health [26, 27], which are often based on measures of the richness and abundance of indicator groups as in the many variants of the widely used Index of Biotic Integrity developed by Karr [12]. Disparities may occur however where microbial richness may remain high in streams heavily modified by human activities [28], while benthic macroinvertebrates and fishes commonly decrease to a few tolerant taxa [29, 30]. Within the microbial metacommunity, the high OTU (Operational Taxonomic Unit) richness and dispersal from multiple sources [31, 32] can maintain diversity during environmental change, while many eukaryotes have a substantially more limited species pool and typically show more limited dispersal [33, 34]. The high expected turnover of microbial diversity and OTU composition may provide additional opportunities and temporal sensitivity to assess recent changes in ecosystem status, even in streams where the legacy of previous disturbances may have eliminated other eukaryotic indicators that are not quick to recolonize.

Despite the potential of microbe-based assessment methods for assessing stream condition, important questions must be answered before their widespread adoption. Among the

hundreds of OTUs present within a single sample [35], some should respond strongly to specific stressors and could serve as indicator taxa of ecosystem health, but we know little about their specific responses, ubiquity, behavior, or strength of relationship with environmental gradients. The appropriate level of taxonomic resolution, or aggregation, also remains in question. Most OTUs in environmental samples remain unclassified, whereas coarsening the resolution to include more classified microbes (e.g., class-level) may dilute information from specific OTU indicators because of other related taxa in the sample. In contrast, use of OTU-level data for developing bioindicators may suffer from over-fitting so that every sample represents a unique community, potentially leading to model predictions so specific that they show poor transferability when applied outside of their local watershed. Identifying the appropriate taxonomic resolution is therefore a central question for assessing the usefulness of microbes as bioindicators of stream condition. Finally, the most effective type of environmental sample and when to collect it remain undecided. To date, most stream microbial research has focused on biofilms scraped from rocks [e.g., 26, 36], but samples collected from stream sediments or directly from the water column might provide complementary or better insights. For example, Hosen et al. [25] suggested that water column bacterial communities respond more strongly to watershed urbanization, whereas communities in stream sediment samples better reflected environmental conditions within the sampling reach. A better understanding of these response differences might allow for more refined predictions, and hence could facilitate broader application.

In this paper, we analyze high-throughput sequencing data microbial 16S diversity from a watershed-scale sample to show how the stream microbes can be effectively used to characterize the ecological condition of freshwater streams and how the results compare with benthic macroinvertebrates, a traditional taxon used for assessment. We develop a methodology based on DAPC (Discriminant Analysis of Principal Components) that is used widely in population genetics research [37], but is not traditionally applied to microbial analyses. We also evaluate the influence of taxonomic aggregation on stream condition classification. Finally, we use machine-learning models of community compositional turnover (Gradient Forest [38]) to associate microbial taxa and community-level relationships to a suite of potential stressors known to alter stream ecological condition.

## Materials and methods

During 2014 we sampled 82 headwater streams (1st-3rd order) in Maryland across diverse gradients of geography and land use at two timepoints—spring and summer (Fig 1). Maryland geography changes on a west to east gradient from its mountainous highlands (a conglomeration of Appalachian Plateau, Ridge and Valley, and Blue Ridge physiographic provinces) to rolling Eastern Piedmont to the Atlantic Coastal Plain where the land meets tidal areas. Watershed land uses ranged from almost entirely forested to those dominated by agriculture (max = 73%) or urban uses (max = 91%).

Each sampling site was co-located with sampling carried out by the Maryland Biological Stream Survey (MBSS). The MBSS program is the principal monitoring and assessment program for nontidal streams in the state. MBSS sampling includes benthic macroinvertebrates and water chemistry in a spring sample and fish and physical habitat in a summer sample [3]. Organismal collections are used to assess each stream with a Benthic Index of Biotic Integrity (BIBI) specifically developed for benthic macroinvertebrates and a Fish Index of Biotic Integrity (FIBI) for fishes. Both the FIBI and the BIBI for Maryland have been well documented [39] and are legal biocriteria with the ability to trigger a stream's placement on the 303b impaired waters list within the state. Scoring for both the FIBI and the BIBI is continuous

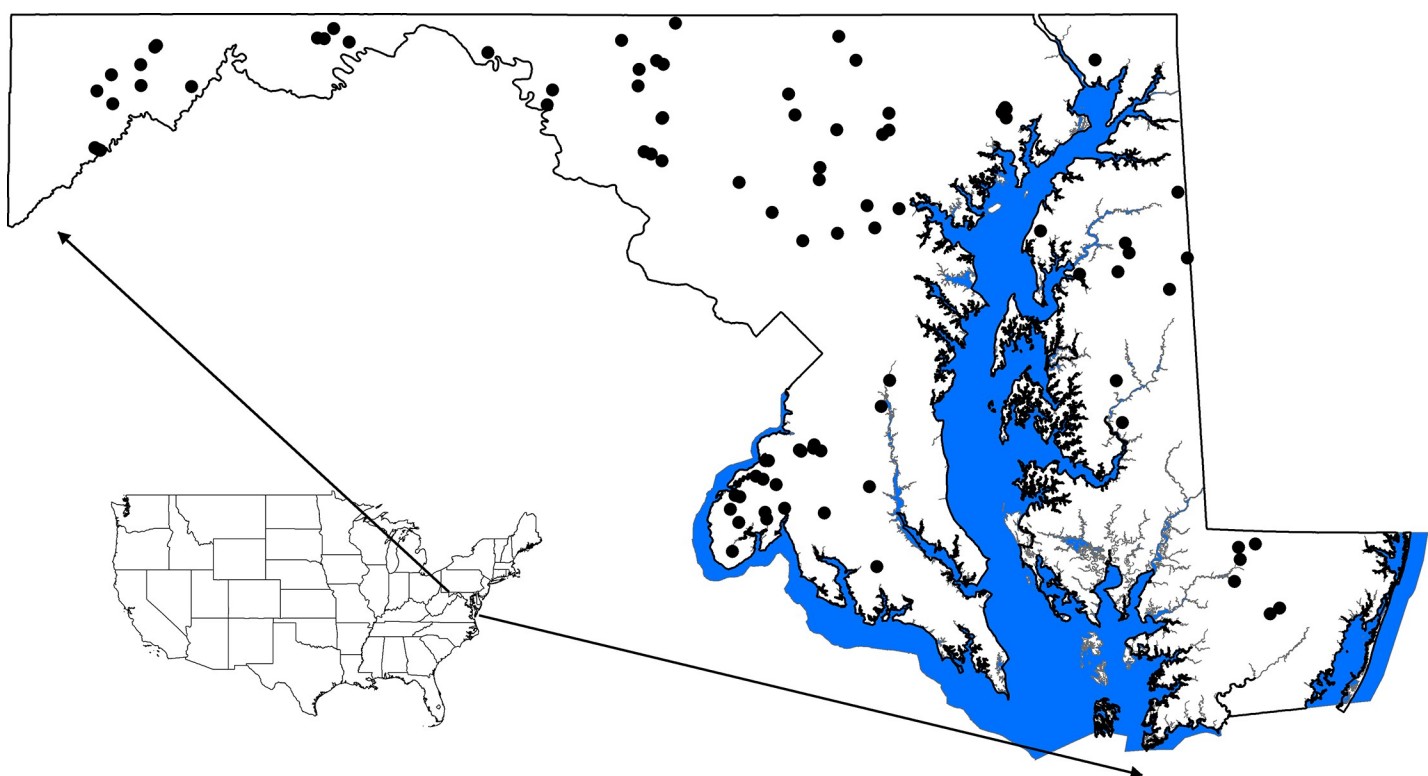

**Fig 1. Map of the study area.** We collected water column and sediment microbial samples in both spring and summer in 82 streams across the state of Maryland, USA. Nation and state outline source: US Census Bureau 2018 TIGER/Line Shapefile. Coastal boundary source: Maryland Geological Survey.

within a range from 1 (very poor) to 5 (very good). Across the 82 streams sampled for analyses, five streams had BIBI scores rating as very poor with BIBI between 1 and 2; 18 rated poor (BIBI between 2 and 3; 23 rated as fair (BIBI between 3 and 4), and 37 sites rated as good with a BIBI ranging between 4 and 5.

At each site, we collected two types of samples to characterize the stream microbial community: water and sediment. Water samples were collected in sterilized 0.5 L bottles and placed immediately on ice and into refrigerated storage within 24 hours of collection. We collected sediment samples by plunging the wide end of a 60cc sterile syringe into the first 1cm of stream sediment, inserting the syringe plunger, capping the small end, and placing the entire sample into a sterile 1L Whirlpak before storage in refrigeration and transport to the laboratory. Thus, the sediment samples are comprised of both surface biofilms and sub-surface microbes. Sediment samples were all collected in shallow depositional areas, commonly from the lower end of glides between the thalweg and stream margin. This permitted consistent coring within depositional silts and sands without involving significant amounts of leaf material and other detritus that accumulates in the bottoms of pools. Sediment samples were stored at -80˚C until processing. No permits were required for collecting either the water or sediment samples. Permissions were obtained by the MBSS from each land owner prior to accessing sampling sites. Access to the monitoring data was granted by the MBSS program. The sampling area was bounded by the range 39.72N, 79.47W and 38.02N, 75.35W.

Water samples were maintained in refrigerated storage until being filtered within 4 days of collection. 500ml samples were vacuum filtered through a 0.22 μm pore size, 47mm diameter, polyethersulfone filter. After filtration, each filter was aseptically quartered, transferred to a sterile centrifuge tube, and stored at -80˚C until time of DNA extraction.

## DNA extraction and library preparation

DNA was extracted from the water filters and sediment cores using the MoBio PowerSoil–htp 96 well Soil DNA Isolation Kit with modifications. For the water samples, two quarters of each filter were placed in a 5 ml MoBio PowerWater bead tube and 925 μl of PowerSoil-htp bead solution and added 75ul of solution C1, a proprietary component of the MoBiop PowerSoil-htp extraction kit and used to aid in cell lysis during bead beating. These volumes were adjusted in order to maintain the appropriate ratio as communicated by MoBio. Samples were then vortexed for a total of 10 minutes at maximum speed. After vortexing, 20 μl of Proteinase K (20 mg/ml) was added and samples incubated at 56° C for 30 minutes. The samples were then centrifuged for 1 minute at 3000xg. Approximately 500 μl of each supernatant was transferred to a 96 well plate.

Three sediment cores were obtained per site per season except for sites JONE315, LOCH120, JONE109, RKGR119, and LIBE102 where eight cores were obtained for the spring and summer samples. Each of these cores was treated as an individual sample during the DNA extraction process and then pooled after the first amplicon PCR. For each core, the sediment was transferred from the syringe to Whirlpak, homogenized, and approximately 250 mg of sample was aseptically aliquoted to a plate well. Each well received 750 μl of bead solution and 60 μl of solution C1. Then the plate was attached to a Qiagen TissueLyser II and shaken at 20 Hz for 10 minutes. The plates were reoriented and the process repeated. After bead beating, 20 μl of Proteinase K (20 mg/ml) was added and samples incubated at 56C for 30 minutes. From this point on the MoBio protocol was followed without additional modifications for both the water and sediment samples.

Library preparation followed the Illumina 16S Metagenomic protocol, with modifications. For amplicon PCR, the primers used were U515F and 806R, which amplify an approximately 250bp region of V4 of the bacterial and archaeal 16S subunit ribosomal gene. The primers consisted of an Illumina overhang adapter sequence as well as a locus-specific sequence (in bold-face below): Forward primer 5' TCGTCGGCAGCGTCAGATGTGTATAAGAGACAG**GTGCCAGC MGCCGCGGTAA** 3' and reverse primer 5' GTCTCGTGGGCTCGGAGATGTGTATAAGAGA CAG**GGACTACHVGGGTWTCTAAT** 3'.

Each PCR reaction consisted of 2.5 μl of extracted DNA (concentration ranged from approximately <1ng/ul to >10ng/ul), 5 μl of the forward and reverse amplicon primers (each at a stock concentration of 1uM); 12.5ul of 2x KAPA HiFi HotStart Ready Mix for a total volume of 25ul. PCR was run on an Eppendorf thermocycler with parameters of 95C for 3 minutes followed by 25 cycles of 95° C for 30 seconds, 65° C for 30 seconds, 72° C for 30 seconds, a final extension step of 72° C for 5 minutes and a hold at 4° C. 5ul from a random sample of PCR products were run on a 1.5% agarose gel to verify the expected size of ~350bp (amplicon with primers). Two negative controls and one positive control (Microbial Mock Community B, catalog number HM-276D, BEI resources) were included. The sediment PCR products corresponding to a particular site and season were then pooled and 25ul of each pooled product transferred to a new 96-well plate before clean-up. We pooled PCR products to avoid a scaling mismatch because collections at a site and season were subsamples within a reach; the MBSS benthic macroinvertebrate samples are also pooled from several subsamples throughout the same reach.

PCR products were cleaned using 20 μl of AMPure XP beads per 25 μl of PCR product following the manufacturer's instructions, and resuspended in 50 μl of 10 mM Tris pH 8.5 buffer. Individual samples were then barcoded in an indexing PCR reaction, consisting of 15ul cleaned PCR product; 25 μl of 2x KAPA HiFi HotStart Ready Mix; 5 μl each of Nextera XT Index 1 Primers (N7XX) and Index 2 Primers (S5XX) from the Nextera XT Index kit for a

total volume of 50 μl. Primer sequences can be found in the Illumina documentation: https://support.illumina.com/content/dam/illumina-support/documents/documentation/chemistry_documentation/experiment-design/illumina-adapter-sequences-1000000002694-12.pdf. PCR parameters were 95˚ C for 3 minutes; 8 cycles of 95˚ C for 30 seconds; 55˚ C for 30 seconds; 72˚ C for 30 seconds; a final extension of 72˚ C for 5 minutes followed by a 4˚ C hold. The PCR index product was cleaned as described above with the following modifications: 56 μl of AMPure XP beads was added to each Index PCR product and eluted in 25 μl of 10 mM Tris pH 8.5 buffer. Each sample was quantified using the Qubit dsDNA High Sensitivity Assay Kit, and then shipped to the University of Maryland Center for Environmental Science-Institute of Marine and Environmental Technology for normalization and sequencing (150 bp paired-end reads) on an Illumina MiSeq.

Sequence data are available from NCBI under BioProject accession number PRJNA545742.

## Bioinformatic analysis

All bioinformatic analyses were carried out using the *mothur* software package (v. 1.31.2) [40]. Sequences were trimmed according to base quality scores using a 50 bp sliding window with an average quality score cutoff of 35. Reads with primer mismatches, ambiguous bases, homopolymers greater than 8 bp, and/or sequences less than 100 bp and greater than 250 bp were removed. Due to inadequate read overlap after trimming, only the forward reads were used for analyses. Sequences were aligned to a reference alignment (SILVA v. 119), and sequences that did not align were removed. All remaining sequences were trimmed to the same start and end position, and any unnecessary gaps generated during alignment were removed. Prior to chimera removal, sequences were preclustered allowing a difference of up to 2 bp (1.5%) between sequences. Chimeras were detected using the UCHIME algorithm [41] and removed from the dataset. Sequences were classified using a Bayesian classifier with an 80% pseudobootstrap confidence score against the GreenGenes database (v. 13.8.99). All sequences that classified as unknown, chloroplasts, mitochondria, or *Eukaryota* were removed. Prior to clustering the sequences into operational taxonomic units (OTUs), sequences were split into bins based on their taxonomic order to reduce the computational demands of clustering. OTUs were clustered at a 3% dissimilarity level using an average neighbor algorithm and normalized to 1344 sequences. The error rate of the sequencing was determined using our positive control Microbial Mock Community B and default parameters of the command seq.error.

## Statistical analysis

We used Discriminant Analysis of Principal Components (DAPC) as implemented in the *adegenet* [42, 43] package for R [44] to classify and predict the ecological condition of headwater streams based on archaea and bacteria OTU identities.

DAPC was initially developed for population genetic processing of thousands of single nucleotide polymorphisms (SNPs) to identify loci that may be associated with specified sample groups. Here, we use it in an analogous context, but not for SNPs within individuals. Rather, we treat OTUs as the equivalent of SNPs and measures of stream condition as the sample groups or traits of interest. Specifically, we used DAPC to identify potential OTUs that are associated with stream ecological condition. The approach first performs a principal components analysis on the normalized OTU data to formulate the principal components accounting for the variation in the OTU dataset. The principal components are then used as inputs into a linear discriminant analysis to predict, in our case, a site's membership as specified *a priori* by the BIBI condition score calculated by the MBSS program. Potential microbial indicator taxa for BIBI condition can then be identified by calculating the loadings of each OTU on the

discriminant axes, and retaining taxa that exceed a user-defined threshold. We retained enough principal components to capture 95% of the variation in the OTU dataset for each combination of sampling period (spring or summer) and sampling medium (water or sediment). For the Discriminant Analysis portion of the DAPC, we retained all available axes for prediction to 8 BIBI categories created by subdividing the BIBI, which ranges continuously from 1–5, into eight 0.5 unit categories. We ran analyses separately based on phylum, class, order, family, genus and OTU (species) in order to identify the level of taxonomic resolution that best captured the ability to classify stream ecological condition.

We used a resampling approach to evaluate model performance. For each of the various combinations of taxonomic level, sampling period, and sampling media, 1,000 iterations of the DAPC analysis were performed, with 80% of the dataset randomly selected and used for model building and the remaining 20% used for validation. We report the mean classification accuracies and 95% confidence intervals of each model classification scenario in the results.

Subdividing the continuous BIBI into discrete, non-overlapping categories for the discriminant analyses introduces potential error in the classifications. For example, a difference between a BIBI of 3.9 and 4.1 is small, yet the two scores fall into different categories. Therefore, we assessed classification accuracy in multiple ways. The first assessment used the accuracy as returned by the DAPC using a leave-one-out approach. The second accuracy assessment allowed for variation by considering a classification as correct if the DAPC classified a sample into the observed BIBI category for that site, or into an adjacent BIBI category (i.e., relaxing the definition of correct assignment to +/- 1 BIBI category). Thus, a DAPC prediction of category 3.51–4.0 would be considered correct if the original BIBI score for a site ranged anywhere from 3.1 to 4.5. We term this "fuzzy accuracy" as it is a form of fuzzy logic in assessing the overall accuracy to deal with uncertainties introduced in moving from a continuous to discrete classification variable. Lastly, we calculated Cohen's Kappa [45, 46] for rater agreement of the cross-classification to test whether the overall accuracy or fuzzy accuracy was significantly different from random guessing in order to account for the non-uniform distribution of BIBI scores across sites.

To determine the relative importance that a smaller set of microbial indicator taxa may contribute to the classification of stream BIBI condition, we identified the taxa most important in predicting stream condition in the DAPC, and re-assessed the classification accuracy using only these taxa. Important taxa were identified as having loadings of 0.03 or greater on one of the three DAPC discriminant axes. These important taxa were then subset from the larger dataset and used to predict a stream's condition as specified *a priori* by the BIBI. Thus, we re-ran the DAPC resampling, but restricted it to the small subset of important taxa—typically 15–30 taxa depending on taxonomic resolution. We assessed both the leave-one-out accuracy and the fuzzy accuracy on both the training and the validation datasets using the resampling approach described previously.

Lastly we used the machine-learning algorithm Gradient Forest (GF) [38] to identify important environmental variables contributing to microbial community composition, and the association of particular indicator taxa with these variables. The GF analysis is thus complementary to the DAPC analysis, as it helps to identify which of the environmental predictors is most important in explaining the among-site variance in abundance for taxa identified as important indicators in the DAPC analysis. GF also gives insight into the shape of the response between important taxa and environmental gradients (e.g., linear vs. threshold effects). We used relative taxa abundance for each site as the response variable, predicted by a multivariate set of environmental predictors for each site that include chemical attributes, physical attributes, and upstream catchment land uses (Table 1). Each GF model consisted of 500 bootstrapped regression trees. The importance of correlated environmental predictors was assessed via conditional permutation for predictors correlated above $r = 0.5$ (see also [38]).

**Table 1. Environmental variables used in the gradient forest analysis.**

| Variable | Abbreviation | Definition |
|---|---|---|
| **Chemical** | | |
| pH | pH | |
| Acid Neutralizing Capacity (mg/L) | ANC | |
| Dissolved Organic Carbon (mg/L) | DOC | |
| Total Nitrogen (mg/L) | TN | |
| Nitrate Nitrogen (mg/L) | NO3 | |
| Ammonia (mg/L) | NH3 | |
| Total Phosphorus (mg/L) | TP | |
| Orthophosphate (mg/L) | OPhos | |
| Chloride (mg/L) | Cl | |
| Sulfate (mg/L) | SO4 | |
| Specific Conductance (μS/cm) | Cond | |
| Carbon (%) | Carbon | Percent of particulate carbon in sediment samples |
| **Landscape** | | |
| Forest Cover (%) | Forest | Forested land in the upstream catchment |
| Urban Cover (%) | Urban | Urbanized land in the upstream catchment |
| Agriculture Cover (%) | Agriculture | Agricultural land in the upstream catchment |
| Impervious Surface Cover (%) | ISC | Impervious surfaces in the upstream catchment |
| Human Disturbed (%) | Human | Urban + Ag land covers in the upstream catchment |
| Catchment Area (ha) | Area | Area of the upstream catchment |
| **Physical** | | |
| Embeddedness (%) | Embed | Fine sediments covering the stream bottom |
| Eipfaunal Substrate (1–20) | Epi | Amount and quality of hard surfaces |

# Results

The normalized dataset contained 61,172 OTUs, with roughly 88% of sequences classified for phyla, 73% to order, 57% to family, 32% to genus, and less than 1% to OTU (Table 2). Bacteria taxonomic richness was always at least an order of magnitude greater than for archaea richness within each taxonomic level (Table 2). At the genus level, we found 634 named genera (19 archaea and 615 bacteria) with all occurring across both water and sediment samples collected during both spring and summer. Unsurprisingly, the percentages of unclassified taxa decreased as taxonomic resolution decreased. Except for OTU-level or as otherwise indicated, all analyses were restricted to classified taxa within a taxonomic level.

Across both seasons and substrate, Proteobacteria dominated samples. Other commonly occurring phyla were *Bacteroidetes* and *Actinobacteria* in water samples, while *Acidobacteria*, *Verrucomicrobia*, *Planctomycetes*, *Chloroflexi*, *Crenarchaeota*, and *Nitrospirae* were common in sediments (S1 Fig). An in-depth analysis of community richness and diversity of these data in the context of landscape setting is found in [47] and is not the intent of the current study. Rather, we describe community representation and responses at the order-level after demonstrating its usefulness in predicting stream condition.

**Table 2. Numbers of classified lineages within each taxonomic level used in the DAPC analyses.** The numbers of unclassified lineages are in parentheses.

| | Phylum | Class | Order | Family | Genus | OTU |
|---|---|---|---|---|---|---|
| Archaea | 3 (0) | 10 (1) | 16 (2) | 16 (11) | 19 (31) | 12 (1129) |
| Bacteria | 64 (9) | 163 (38) | 262 (102) | 319 (241) | 615 (1307) | 593 (59438) |

## Spring and summer sediments predict ecological condition

DAPC analyses of stream microbial data successfully classified the ecological condition of headwater streams that were previously classified by benthic macroinvertebrates with the BIBI. Classification accuracy within the training dataset ranged from 31% to 100% across the various levels of taxonomic resolution, seasons, and sampling media (Fig 2A). Models using the full OTU dataset produced the highest classification accuracies across the different types of

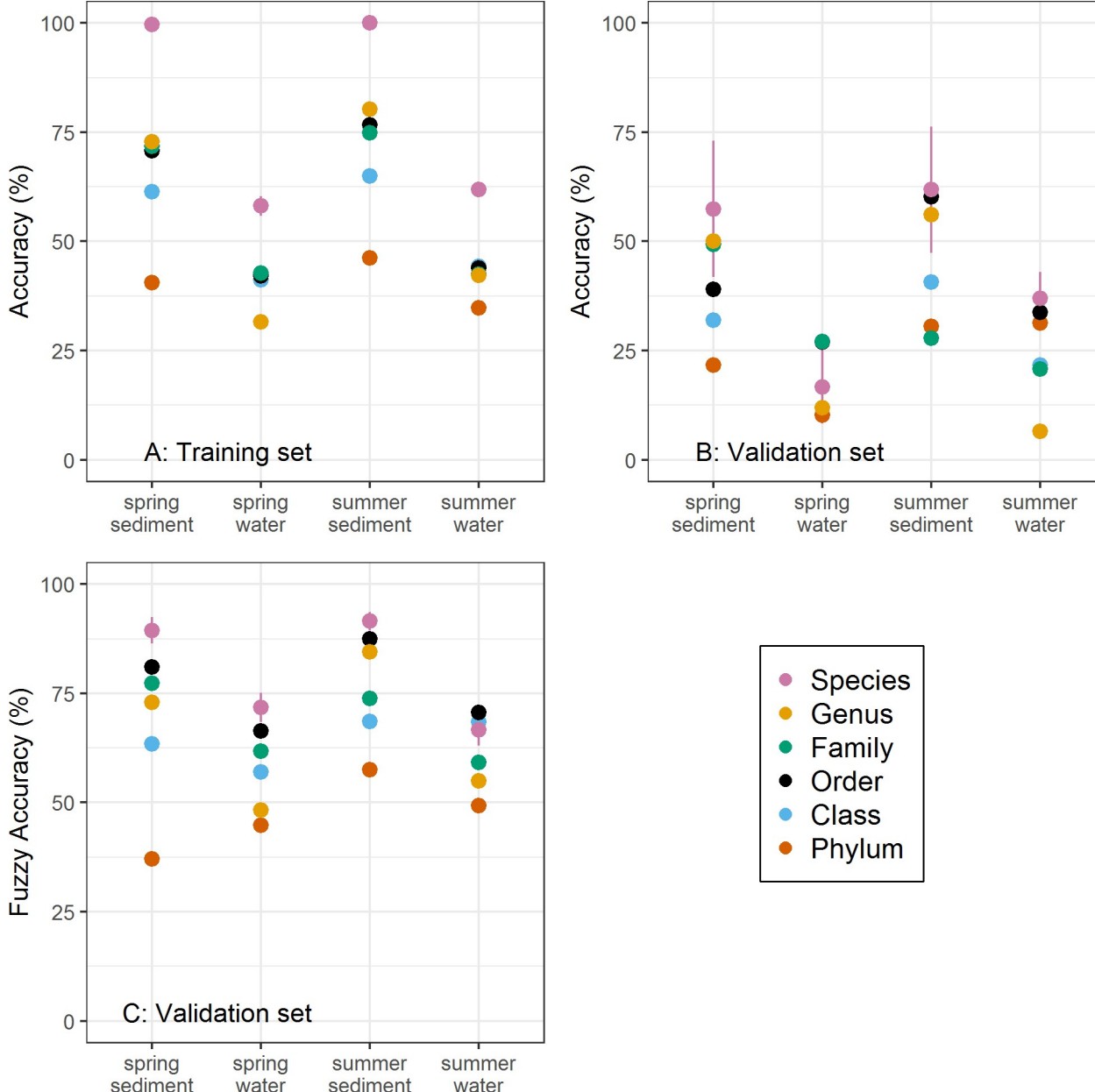

**Fig 2. Stream assessment classification accuracy using all taxa.** Model classification accuracy of sites into 8 possible BIBI categories for the training dataset (Panel A), validation dataset (Panel B), and the fuzzy classification accuracy for the validation dataset (Panel C) across the differing levels of taxonomic resolution, sampling season (spring, summer), and sampling medium (water, stream sediment). Points represent the mean accuracy across 1,000 model runs resampling 80% of the dataset to form the training dataset and the remaining 20% comprising the validation dataset. Results for species are based on all OTUs, including those not classified.

samples. However, coarser levels such as order and genus produced similar accuracies when applied to the validation data. The absolute classification accuracy applied to the validation data was lower than for the training data (Fig 2B). However, most misclassifications fell into an adjacent category, such that the fuzzy classification accuracy of the validation dataset exceeded 80% in some model scenarios including genus- and order-level classifications for summer sediments (Fig 2C).

Sediment samples were considerably better for classifying stream ecological condition in the DAPC analysis than water column samples. Within a season, classification accuracy from sediments typically ranged at least 10–20 percentage points higher than did samples from water for accuracy of both the overall and fuzzy classifications for a given taxonomic level (Fig 2). The differences in classification accuracy between sediment and water samples were generally lowest at the coarsest taxonomic levels (Phylum and Class) and became more pronounced for taxonomic resolutions of order and finer.

Summer sediment samples generally produced higher classification accuracy within a taxonomic level for both the training and validation data compared to spring sediments samples. However, predictions from spring sediments often performed similarly (Fig 2) and suggest that models from sediment samples taken at either of these times may provide good performance.

## A subset of orders predicts ecological condition

Classifications based on just the subset of the most important taxa in the DAPC analysis were less accurate than analyses using all taxa, yet many scenarios remained significantly better than random for classifications applied to the training dataset (Fig 3A). Although the overall accuracy in classifying streams to their BIBI scores was very low when applied to the validation dataset (Fig 3B), the accuracy based on the fuzzy classification was often high (Fig 3C) and approaching those when using all taxa within a level (Fig 2C). For example, the average fuzzy classification accuracy in summer sediments using only the important taxa exceeded 75% for both order- and genus-level analyses (Fig 3C).

Classification accuracies based on the subset of important OTUs often performed less well than models built with coarser taxonomic resolutions, contrasting with results based on all-taxa models. Order-level taxa generally performed better than other taxonomic levels when applied to the validation dataset, including the OTU-level. For example, classifications using only 31 orders in summer sediments and 29 orders in spring sediments (S2 Fig) produced fuzzy accuracies above 75% (Fig 3C). Most of these orders were ubiquitous, occurring in at least 90% of the samples. In addition to orders, fuzzy classification accuracies were also reasonably high when using genus- and family-level taxa for both spring and summer sediments.

Almost all of the orders important in classifying stream condition were also among the most numerically abundant in sediment samples. In particular, *Rhizobiales*, *Chthoniobacteriales*, *Pirellulales*, *iii1-15*, and *Pedosphaerales* were all highly abundant, ubiquitous, and important to predicting stream condition in both spring and summer sediment samples (Fig 4). Some orders such as *NRP-J*, *GCA004*, and *Thermogemmatisporales* were quite important for prediction, but were not among the 25 most abundant orders. These taxa also tended to be less ubiquitous in occurrences across samples (S2 Fig). Aside from *Actinomycetales*, *Flavobacteriales*, and *Burkholderales*, important orders were not as abundant in water column samples and demonstrate differences between the sediment and water column communities.

## Community turnover relates to water chemistry

Based on the DAPC analysis which showed high classification accuracies at the order level and finer taxonomic resolutions, we focused the Gradient Forest (GF) community turnover

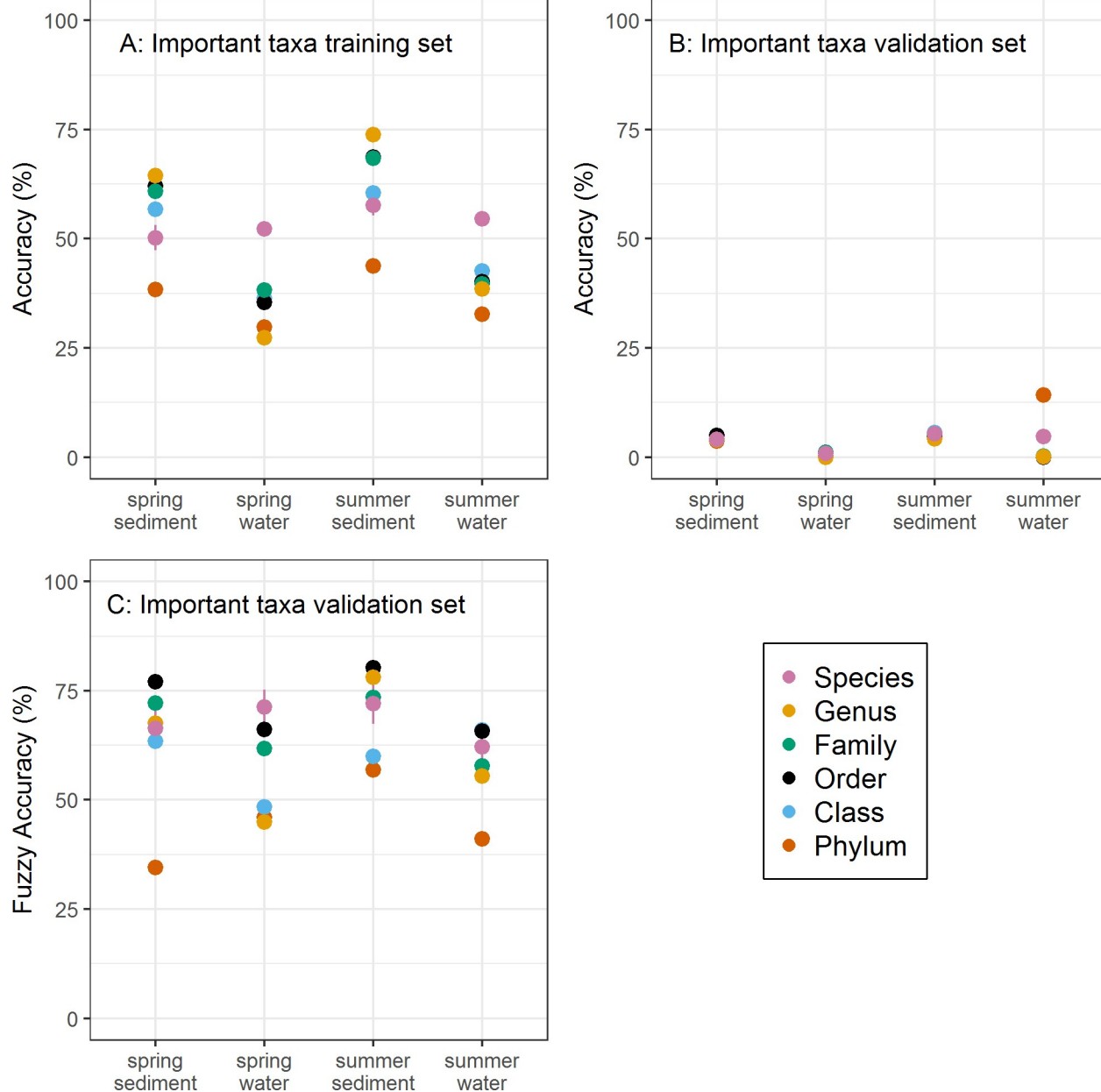

**Fig 3. Stream assessment classification accuracy with a subset of important taxa.** Model classification accuracy of sites into 8 possible BIBI categories based on models using only those taxa loading heavily (0.03 or greater) in the DAPC analysis. Accuracies are shown for the training dataset (A), validation dataset (B), and the fuzzy classification accuracy for the validation dataset (C) across the differing levels of taxonomic resolution, sampling season (spring, summer), and sampling substrate (water, stream sediment). Points represent the mean accuracy of 1,000 model runs resampling 80% of the dataset to form the training dataset and the remaining 20% comprising the validation dataset.

analyses on order-level taxa. Using GF, we identified several environmental gradients associated with the order-level community compositional turnover. The most important environmental predictors associated with community turnover were shared between spring and summer microbial communities, and included pH, acid neutralizing capacity, dissolved organic carbon, nitrate nitrogen, the percent of particulate carbon present (for sediments) in

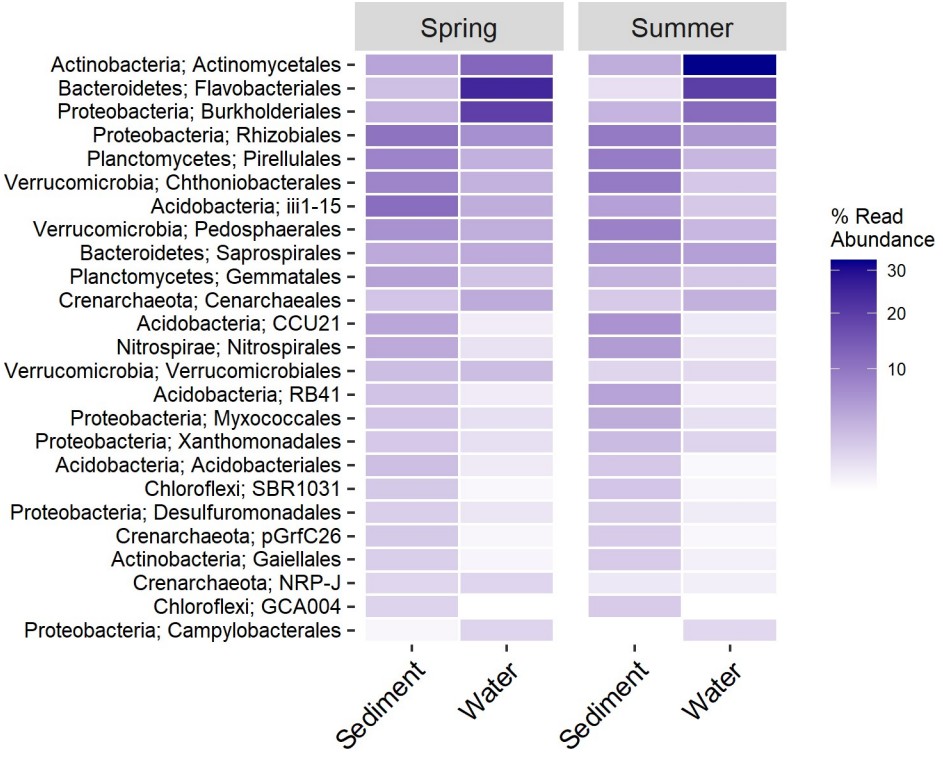

**Fig 4. Abundances of orders important to predicting stream condition from sediment samples.** Taxa are arranged top to bottom by their overall abundances within the entire dataset including water sample.

the sample (Fig 5). In addition to those chemical attributes, embeddedness (the amount of fine sediment deposited throughout the stream reach from which the sample was collected) was also important. The remaining predictors contributed little to explaining among-site variation in community composition, including the various broad land use categories. The most important environmental predictors associated with community turnover were similar between analyses using all named orders and those using just the subset of orders important in the DAPC analyses for both spring and summer sediment communities (Fig 5).

The orders important in the DAPC analysis had variable responses to the most important environmental gradients. Far fewer than half of the 29 taxa typically responded strongly to any given environmental variable, and the responses were usually not concentrated at one specific point along each gradient (S3–S5 Figs). For example, *Pedosphaerales*, *Acidobacteriales*, and *Elin6513* all responded strongly to pH, but at different values, whereas other taxa responded in less dramatic fashion (Fig 6A). Thus, the aggregate community response for each variable was a relatively smooth curve above the lowest pH values (Fig 6B) rather than a stepped-threshold response that would indicate an abrupt transition to a different community. We found similar response gradients of taxa to the overall BIBI score as well as to different land use gradients within the watersheds (S4 Fig), but none of them were as important to community turnover as were many of the chemical variables. *Acidobacteriales* and *Actinomycetales* were strong indicators of very low BIBI scores at a site, whereas *Xanthomonadales* and *Chthoniobacterales* indicated progressively higher BIBI scores, respectively. Similarly, iii1-15 and RB41 were strong indicators of increased urbanization (S4 Fig). Despite these orders being strong indicators, many were present in nearly every sample (S2 Fig), and so the relative abundances or ratios define the response rather than presence or absence in a sample.

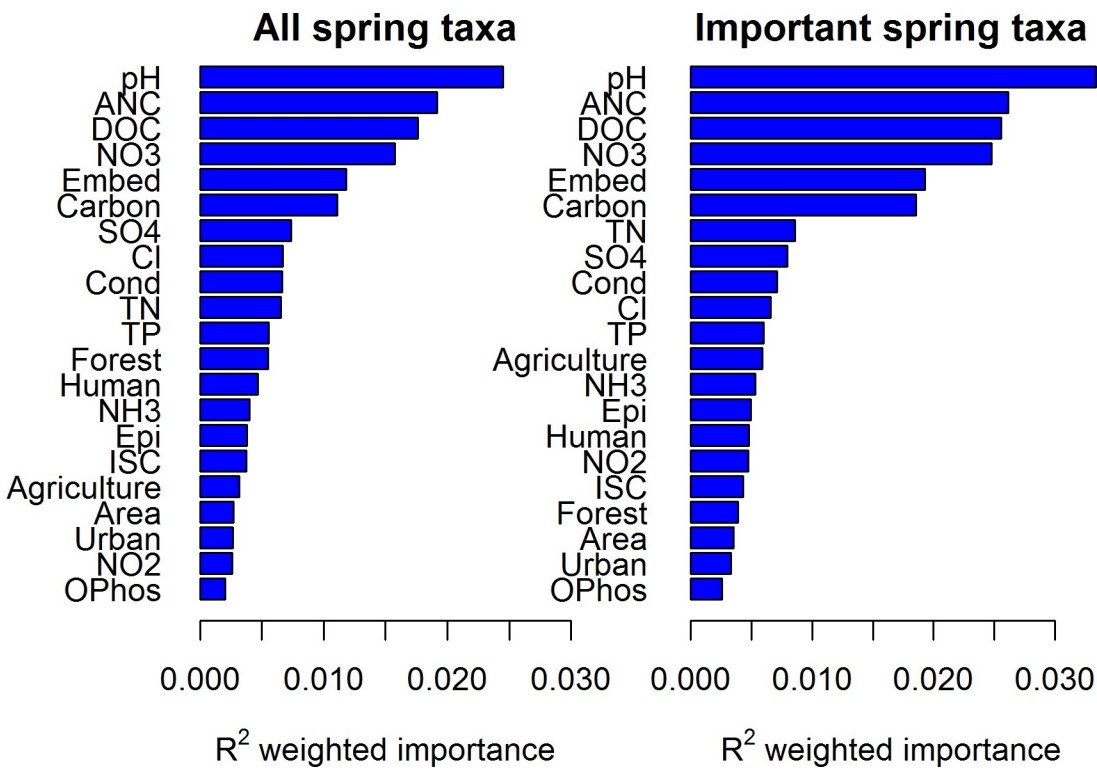

**Fig 5. Importance of environmental variables.** Importance rankings of environmental variables to order-level taxa in spring-collected sediment samples as determined by gradient forest. The left panel shows environmental variable rankings with respect to all taxa, while the right panel does the same for the 29 important order-level taxa. The summer sediment microbial community behaved similarly and is excluded for brevity. Variable definitions can be found in Table 1.

## Discussion

We were successful in using stream sediment microbial communities to reproduce the results of an alternative stream assessment method based on benthic macroinvertebrates. Maryland's BIBI is a well-established assessment method [39] and has legal biocriteria within the state for evaluating the ecological status of small streams. Previous have attempted to relate microbial communities with stream ecosystem status with mixed results. Although Lear et al. [48] could only distinguish the most degraded sites with bacterial biofilms, more recent work has refined the correlations between microbes and stream macroinvertebrate-based assessments [26] and linkages with urban streams [27]. These advances and the current study reflect the large potential of microbial approaches for assessing ecosystem status.

The classification accuracies of the DAPC model compared favorably to accuracies of BIBI scores calculated on replicate benthic macroinvertebrate samples collected by the MBSS program. For field replicate samples collected in 55 randomly selected reaches by the MBSS program (tables 9–10 [49]), we applied the same number of categories and cutoffs to the BIBI as we did in assessing the DAPC model accuracies. The overall accuracy calculated for the MBSS BIBI replicates was 42% and the fuzzy accuracy was 87%. Thus, two benthic macroinvertebrate samples collected at the same time and in the same location had differences in their calculated BIBI scores. Our classification accuracy on the validation dataset had comparable accuracy to the BIBI scores in spring and summer sediment samples for the all-OTU models (57% and 62% overall and 89% and 92% for fuzzy) and the all-orders models (39% and 60% overall and 81% and 88% for fuzzy). Surprisingly, the subset of important orders performed similarly well

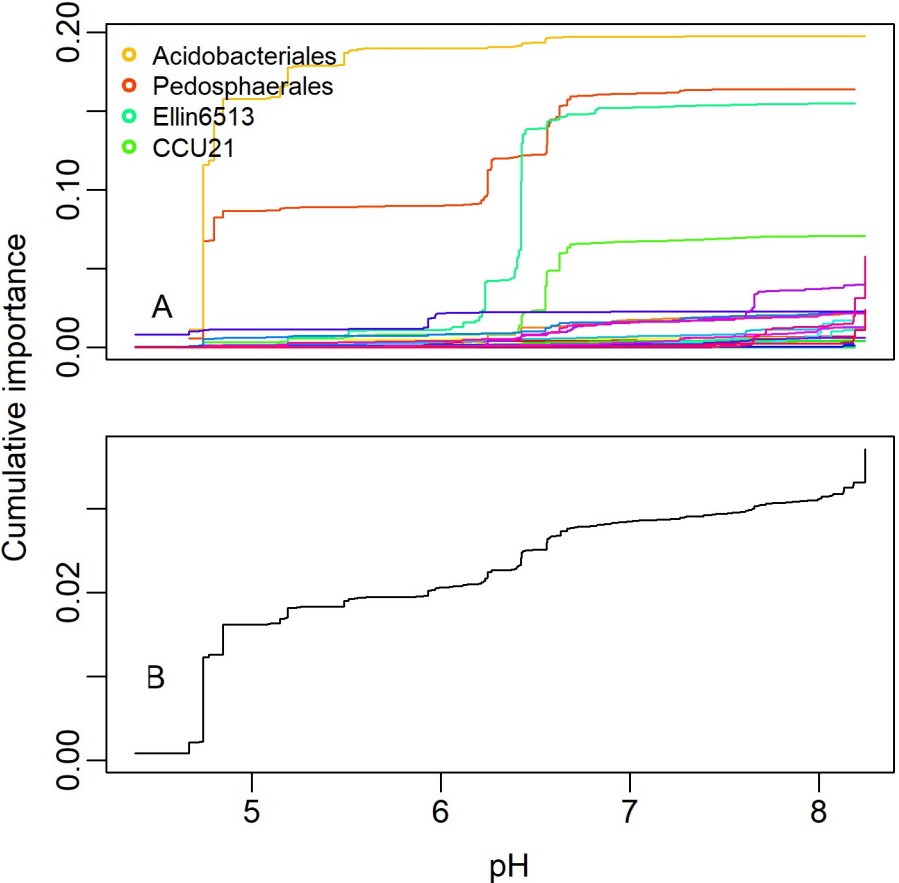

**Fig 6. Taxon-specific responses of the important orders in spring sediment samples along the pH gradient in the gradient forest analysis.** Each line represents a different microbial order.

(77% and 80% for fuzzy). The limited subset's ability to predict the BIBI greatly extends the utility of the approach. While models using all OTUs require all information to be present for modeling, the subset of known indicator taxa can be used for future assessments on streams outside of the study without the need to sample the entire community anew across multiple streams to build a training set. Including all OTUs often produced the best results, but the OTU-level models appeared to be overfit as their accuracy decreased relative to other taxonomic levels when applied to the validation dataset. The decreased performance likely reflects heterogeneous spatial distributions of individual OTUs that disappear as these become aggregated at more coarse levels of taxonomic resolution.

Although family- and possibly genus-level aggregation showed predictive potential, we chose the order-level for multiple reasons. Unclassified OTUs are not uncommon with 16s samples from streams, and aggregation increases the likelihood of including more OTUs within a group for analysis even if these are not classified at the species level. While the DAPC analysis at the species level did include all OTUs, the predictions to the validation data often did not fit as well as the aggregated levels, suggesting model overfitting. We were surprised that order-level aggregation worked well because of the substantial functional differences among members within many orders. However, orders also work well for stream benthic macroinvertebrates where the *Ephemeroptera*, *Plecoptera*, and *Trichoptera* (EPT) form a frequently used diagnostic triad due to their pollution sensitivity. Not all members of EPT are

pollution sensitive, but the aggregated response has more utility than splitting into finer groupings despite genus-level forming the accepted standard resolution. Similarly, a high percentage of *Diptera* is often diagnostic of ecological impairment [39]. Both functional and habitat diversity of members within *Diptera* and each of the EPT are broad [50]. Indeed, large functional diversity may be advantageous by providing a wider range of habitats and conditions for diagnostic groups within an order to survey.

A disadvantage of aggregation is the loss of specificity in understanding the roles of taxa in the environment. Among the many orders important to predicting stream condition, only a few have narrowly described natural histories. *Cenarchaeales* and *Nitrososphaerales* are ammonia oxidizing *Archaea* [51], while the bacteria, *Nitrospirales*, oxidize nitrite-nitrogen [52] and *Desulfuromondales* reduce sulfates [53]. Interestingly, none of these taxa responded strongly to their respective chemical gradients (S4 Fig). However, the methane oxidizing *Methylococcales* (*Protebacteria*) and putative anaerobe *GCA004* (*Chloroflexi*) were strongly related to high levels of embeddedness arising from substantial fine sediment deposition in the stream and associated anoxic environments. Hosen et al. [25] reported a *Methyloccolales* OTU strongly associated with urban streams, but that most methanotrophic or methylotrophic OTUs were more indicative of forested systems, possibly due to deeper groundwater and hyporheic flowpaths that are also anoxic. The remaining important orders are either poorly described or have environmental roles too diverse for general characterization due to their high functional diversity.

Because stream condition can be influenced in numerous ways, a low BIBI score can be achieved by degradation due to urbanization [54], agricultural activities [55], or any number of specific insults such as high sedimentation [56] or altered water chemistry in even forested watersheds [57]. Streams may be influenced simultaneously by multiple sources of degradation, so no single indicator taxon can reflect all potential stressors. Indeed, we found several indicator taxa, but none were specifically diagnostic of overall stream condition because of the many possible ways a stream can become impaired, and most were present in nearly every sample. Rather, the relative proportions of taxa within a community may be a better integrative measure and is similar to what the DAPC analysis accomplished. This is also the general approach of the IBI-based methods that evaluate the overall target community by aggregating and summing numerous indicator metrics [13]. Nonetheless, the environmental gradients measured in our study influenced community turnover, with numerous taxa responding strongly to specific gradients and acting as indicator taxa. The important orders from the DAPC analysis showed similar patterns to the entire community and suggests this subset is both diagnostic of stream condition and representative of the overall microbial sediment community. Numerous indicator taxa correlate strongly both with broad land use categories as well as with specific chemical constituents [25, 27, 47]. Despite strong taxon-specific responses of indicators, we found no community-wide threshold response. Rather, when individual taxon responses were aggregated across the entire community, the result was a fairly smooth and consistent change across each gradient. This lack of abrupt transitions in the microbial community differs from fish and benthic invertebrate communities [58] and may reflect high functional diversity and redundancy in stream sediment microbes, with multiple taxa readily available to increase in abundance as conditions change across a gradient [28].

Classification performance differences between water column and sediment samples may reflect different ecological processes acting on the microbial community and a subsequent scale mismatch with the benthic macroinvertebrates. We had originally expected classifications from water samples to outperform sediment samples because water samples should better integrate upstream catchment characteristics such as land uses that strongly influence fish and benthic invertebrates. However, the sediment microbial community was substantially

better at replicating benthic macroinvertebrate stream condition assessment scores. We believe the sediment microbial community is better spatially and temporally matched with benthic macroinvertebrates. The water column represents the microbial pool available to colonize stream benthic habitats, but the short residence times may not reflect the conditions under which the benthic macroinvertebrates actually exist. Only a subset of microbial taxa survive on the sediment substrates through environmental sorting, and these taxa share more similar environments to the benthic macroinvertebrates. Hosen et al. [25] found that microbes in water column samples were more strongly connected to urbanized watersheds, but that sediment microbial communities were more strongly connected to environmental conditions within the stream reach, which is more similar to what benthic macroinvertebrates experience.

The microbial community appears to respond differently to natural and human-gradients than eukaryotic indicators despite the ability to predict stream condition. Animal taxa show decreasing richness and diversity with subsequent community-level turnover as watersheds lose forest and transition to urban dominated landscapes [27, 59]. In contrast, land uses were not important in microbial community turnover despite its relationship with the BIBI and significant correlations between land uses and the BIBI within our dataset and elsewhere [60]. In fact, the only important environmental variable shared between the important microbial orders and the BIBI was acid neutralizing capacity, which is a measure of a stream's ability to buffer acidity. The microbial community thus appears to measure different aspects influencing stream condition than can be rigorously quantified with the BIBI. We view this as an advantage rather than a weakness because the microbes may be complementary to benthic macroinvertebrates by integrating different environmental attributes influencing the stream ecosystem. Similarly, the relatively smooth community response across the many environmental gradients measured implies high redundancy in both diversity and function as previously suggested by others [48, 27]. The ubiquity and shorter generation times of microbes may also more closely track environmental conditions or be distributed more evenly within a stream than fish or invertebrates, which tend to be spatially heterogeneous [61]. Thus the microbial community may be a better reflection of conditions at the site than the more mobile and longer lived fish and invertebrates, which can disperse to varying degrees [62] and may be limited by dispersal or the species pool [34].

Reliable assessment of stream ecosystem status remains instrumental to many activities, but the results obtained from one taxonomic group are not always congruent with other groups. For example, the MBSS data used in our analyses show a correlation of only 0.32 between the fish IBI and the benthic IBI scores at a site. We chose the BIBI because it receives widespread adoption and use by ecologists and resource managers, and it shows reasonable responses to stressors in Maryland streams [39, 60]. However, we also attempted to use fish data collected at each site in a Fish Index of Biotic Integrity (FIBI) as a potential endpoint in preliminary analyses, but found poor congruence. We suspect the lack of relationship may be due to fish responding more slowly to environmental changes due to substantially longer generation times, difficulty in recolonizing areas after local extirpations and historic contingencies [63], increased species richness near confluences with larger streams [64], and more patchy distributions, all contributing to higher variation in FIBI scores. Both diatoms [65, 66] and salamanders [67] can also be effective groups, but we had no accompanying data for comparisons. Assessments with diatoms often compare similarly to those with benthic macroinvertebrates [68] and may compare similarly with the microbial community due to their shorter generation times and potential ubiquity.

Microbe-based assessment of stream ecological condition extends environmental assessment beyond the use of traditional indicator organisms and has potential advantages. Sample collection for stream sediments is very quick. We frequently spent less than 5 minutes at each

site. The limiting factor is ensuring that all materials are handled and processed to avoid contamination with foreign DNA, particularly from other sites. Collected samples are also easily stored on ice prior to laboratory processing. The rapidity and low-effort for sample collection allows far more samples to be collected in a given day than can be accomplished with fish, salamanders, or benthic macroinvertebrates. A single person can sample many sites in a day with the major limitation being travel time between sites, which makes the overall cost per sample potentially much lower for microbial-based assessment compared to the other approaches. Our results suggest that the microbial communities of spring- or summer-collected stream sediments can reasonably replicate the spring-collected benthic macroinvertebrates that make up the BIBI. Thus, the time window for sampling may be extended far beyond other biological assessment approaches. The microbial community may also allow evaluations of ephemeral streams or those having been restored, but have limited species pools for fish or invertebrates to recolonize. With further research, we view microbes as being a useful coarse filter to cost effectively evaluate many streams and identify those having outlier values (good or bad) to trigger more thorough examinations or possibly to identify candidate streams now suitable for reintroducing extirpated fish or invertebrates.

While the applications of microbial sequencing to stream assessment are evident, many basic questions regarding the processes structuring microbial communities remain unanswered. The overwhelming majority of OTUs and even genera remain unnamed. Our sequencing used single end reads of relatively short length and likely made positive assignments underpowered. Nonetheless, some unclassified taxa appear to be important in classifying stream condition. Whether their importance manifests as indicators, commensals of benthic invertebrate indicators, or as ecosystem engineers remains unknown. Similarly, we know little about the roles of most microbial taxa or the degree of redundancy in ecosystem functions as taxa are aggregated into coarser taxonomic levels, yet microbial activities such as decomposition, nutrient cycling, and respiration are central to stream ecosystem functioning [69]. These activities are undoubtedly influenced by the frequency and magnitude of disturbance as well as the community's response, which remains unknown. As sequencing platforms become more powerful and less expensive, we are excited by the prospects of much deeper basic and applied understanding.

Our findings are an encouraging next step in using the microbial community to better understand stream ecosystem health. However, we believe that more testing is required and that similar efforts need to be replicated in time and across larger areas. Our sampling covered a large geographic area with several physiographic provinces, and we are reasonably confident that the data are representative of a large portion of regional headwater streams. Future research should assess if the same taxa are important for other watersheds both regionally, continentally, and globally.

## Supporting information

**S1 Fig. Abundances of the most common Phyla.**
(TIFF)

**S2 Fig. Important orders.** Frequencies of inclusion in the DAPC analysis as an important order for the most important order-level taxa to predicting stream condition from summer sediment (top panel) and spring sediment (bottom panel) samples. The numbers in parentheses beside each taxon indicate its relative frequency of occurrence at sites in spring or summer sediment samples.
(TIFF)

**S3 Fig. Responses of diagnostic orders to important water chemistry gradients.**
(TIFF)

**S4 Fig. Responses of diagnostic orders to the BIBI and land use gradients.**
(TIFF)

**S5 Fig. Responses of diagnostic orders to water chemistry gradients.**
(TIFF)

## Acknowledgments

The Maryland Biological Stream Survey provided data for benthic macroinvertebrates; fish; physical, chemical, and land use attributes for the streams involved in this study.

## Author Contributions

**Conceptualization:** Robert H. Hilderbrand, Stephen R. Keller, Alyson E. Santoro.

**Data curation:** Alyson E. Santoro.

**Formal analysis:** Robert H. Hilderbrand, Sarah M. Laperriere, Alyson E. Santoro.

**Funding acquisition:** Robert H. Hilderbrand.

**Investigation:** Robert H. Hilderbrand, Stephen R. Keller, Sarah M. Laperriere, Alyson E. Santoro, Jason Cessna, Regina Trott.

**Methodology:** Robert H. Hilderbrand, Stephen R. Keller, Alyson E. Santoro.

**Project administration:** Robert H. Hilderbrand.

**Resources:** Alyson E. Santoro.

**Writing – original draft:** Robert H. Hilderbrand, Stephen R. Keller, Alyson E. Santoro.

**Writing – review & editing:** Robert H. Hilderbrand, Stephen R. Keller, Alyson E. Santoro.

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
