## [Decision Letter · Decision Letter 0]

24 Oct 2019

PONE-D-19-23872

Using the microbiome to assess the ecological condition of headwater streams

PLOS ONE

Dear Dr. Hilderbrand,

Thank you for submitting your manuscript to PLOS ONE. After careful consideration, we feel that it has merit but does not fully meet PLOS ONE’s publication criteria as it currently stands. Therefore, we invite you to submit a revised version of the manuscript that addresses all the points raised by the reviewer. In agreement with this reviewer, I was also very surprized to see that you have pooled your triplicate samples, depriving you of important data. I share also the same astonishment than the reviewer on the very high proportions of unclassified lineages, which raise the question of the accuracy of your sequence assignation.

In addition to the comments of the reviewer, I have also some comments and suggestions about  you manuscript.

1. I regret that absolutely nothing is provided at the beginning of the results on the composition and structure of your microbial communities. Moreover, it would have been interesting to see for example if there were change in the richness/diversity of these communities depending on the ecological status of the sampling sites.

2. I'm very surprising to see that the best results were obtained in your stream assessment classification when considering the order level and also that the depending of the nature of the samples (sediment versus water, suumer versus spring), species, genus or order provide the best result. IThis point need to be discussed.

3. I don't have understood why you choose to look at the Community turnover by taking into account DAPC analysis showing high classification accuracies at the order level knowing that there were less accurate that at genus or species levels?

4. Figure 6 is quite impossible to read. Thnaks to remove the legend from the figure and to change the format of these legends (why o__RB41  and not only RB4?)

5. I found that like for the result part, you must more deeply discussed your microbiological data. For example, nothing is provided on the orders being important in DAPC model knowing numerous papers have been published int he 10 past years on the comparison of microbial communities in more or less polluted trerrestrial or aquatic ecosystems.

We would appreciate receiving your revised manuscript by Dec 08 2019 11:59PM. To enhance the reproducibility of your results, we recommend that if applicable you deposit your laboratory protocols in protocols.io, where a protocol can be assigned its own identifier (DOI) such that it can be cited independently in the future. For instructions see: http://journals.plos.org/plosone/s/submission-guidelines#loc-laboratory-protocols

We look forward to receiving your revised manuscript.

Kind regards,

Jean-François Humbert

Academic Editor

PLOS ONE

**Journal Requirements:**

3. In your Methods section, please provide additional information regarding the permits you obtained for the work. Please ensure you have included the full name of the authority that approved the field site access and, if no permits were required, a brief statement explaining why."""

4. In your methods section please include the geographic coordinates for the data set.

5. We note that  Figure(s) 1 in your submission contain [map/satellite] images which may be copyrighted. All PLOS content is published under the Creative Commons Attribution License (CC BY 4.0), which means that the manuscript, images, and Supporting Information files will be freely available online, and any third party is permitted to access, download, copy, distribute, and use these materials in any way, even commercially, with proper attribution. For these reasons, we cannot publish previously copyrighted maps or satellite images created using proprietary data, such as Google software (Google Maps, Street View, and Earth). For more information, see our copyright guidelines: http://journals.plos.org/plosone/s/licenses-and-copyright.

a) You may seek permission from the original copyright holder of Figure(s) [#] to publish the content specifically under the CC BY 4.0 license.  

**Comments to the Author**

1. Is the manuscript technically sound, and do the data support the conclusions?

Reviewer #1: Partly

2. Has the statistical analysis been performed appropriately and rigorously? 

Reviewer #1: I Don't Know

3. Have the authors made all data underlying the findings in their manuscript fully available?

Reviewer #1: No

4. Is the manuscript presented in an intelligible fashion and written in standard English?

Reviewer #1: Yes

5. Review Comments to the Author

Reviewer #1: Revision on PONE-D-19-23872. This manuscript proposes the microbiome as tool to characterize the ecological condition of freshwater streams after their results comparison with more traditional biotic integrity indexes based on macroinvertebrates. In addition, modelling based on gradient forest was performed to identify the association of particular taxa with environmental conditions. Below I address some questions/suggestions aiming to improve this manuscript.

This study proposes the microbiome as tool to assess the level of disturbance and degradation of freshwater streams. In L 523-526, the authors highlight that the ubiquity and short generation times of microorganisms would permit a more closely track of environmental conditions comparing to macrofauna. However, the proved adaptability of microbes (ex. gene transfer confering resistance) to a range of perturbations is never discussed.

L 128-138. Discriminant Analysis of Principal Components (DAPC) is mostly used in population genetics were sequences are phylogenetically close (same species). To which extent (provide descriptor) this methodological approach is robust in your analysis of community genetics? This need to be discussed in the methods section.

Authors never explain how do they switch from analysing bacteria and archaea separately (Table 2) to mix all together (Figures 2 to 5). Please, explain somewhere in the methods the rationale behind.

L214-216. Why the authors pooled triplicates samples of DNA extraction in sediments? Keeping replicates separately could have helped to correct the imbalance between sites having very poor scores (n = 5) and good scores (n = 37). How this imbalance can further influence in DAPC and GA analyses?

Did you perform DAPC (and further statistical analyses) with even sequence depths among samples? Did you run rarefaction curves prior to DAPC? Were these sequences selected at random?

Accession number of 16S rRNA sequencing results must be supplied in the manuscript.

The extremely low affiliation of OTUs sequences into species (ex. 593 classified against 59438 unclassified) is normal when using this kind of sequencing approach? If this is an optimal result, authors should highlight this in the discussion section. If not, the sequencing analysis should be re-done.

Threshold values considered as acceptable for % accuracy (Figures 2 and 3) and/or R2 (Figure 5) are never discussed in the manuscript. Please provide information in the methods section.

According to Figure 6 results, DOC and carbon had a strong weight in explaining important taxa distribution in spring. However, co-variation between these two environmental variables tends to occur in samples of water. Authors should should these kind of problems in their analysis.

L 478. Many studies have already worked on the biogeographic distribution of microorganisms. A deeper comparison between these studies and the results obtained in the present study would be needed in the discussion section.

L 571-577. The authors should precise that in the case of “functional approaches” metatranscriptomic analyses would be more appropriate than metagenomics analyses.

Minor comments

L39-40. At this step the terms “training data” and “validation data” have not yet been defined. This can difficult the reader understands.

L43-45. This sentence is a bit contradictory to me since the aim of high throughput sequencing is to enhance the depth of analyses in microbial diversity. Rewrite.

L97-98. This statement is not true for micro-eukaryotes such as diatoms. Correct.

L178. Were water samples collected in triplicate?

L180. What is the “C1” solution? Please describe.

L181-182. Did you use a cell disruptor? Please, provide exact speed values. Idem for L 192.

L 222. Not clear to which primers you refer here. Precise.

L344. Accuracy values for Fig 2a are even lower than 43% (ex. genus and phylum level). Correct.

The authors decided to present a selection of their results (ex. Figure 5 and 6) that in my opinion could be placed as supplementary information.

Please correct typing errors on SI units everywhere in the manuscript and especially in the methods section.

6. PLOS authors have the option to publish the peer review history of their article (what does this mean?). If published, this will include your full peer review and any attached files.

Reviewer #1: No

---

## [Author Response · Author response to Decision Letter 0]

7 May 2020

Please see attached file - "Response to Reviewers"

---

## [Decision Letter · Decision Letter 1]

5 Jun 2020

PONE-D-19-23872R1

Microbial Communities Can Predict the Ecological Condition of Headwater Streams

PLOS ONE

Dear Dr. Hilderbrand,

Thank you for submitting your manuscript to PLOS ONE. After careful consideration, we feel that it has merit but does not fully meet PLOS ONE’s publication criteria as it currently stands. Therefore, we invite you to submit a revised version of the manuscript that addresses the point raised by the reviewer of this MS. I agree with him that it should be clarified.

We look forward to receiving your revised manuscript.

Kind regards,

Jean-François Humbert

Academic Editor

PLOS ONE

Reviewers' comments:

Reviewer's Responses to Questions

**Comments to the Author**

1. If the authors have adequately addressed your comments raised in a previous round of review and you feel that this manuscript is now acceptable for publication, you may indicate that here to bypass the “Comments to the Author” section, enter your conflict of interest statement in the “Confidential to Editor” section, and submit your "Accept" recommendation.

Reviewer #1: All comments have been addressed

2. Is the manuscript technically sound, and do the data support the conclusions?

Reviewer #1: Partly

3. Has the statistical analysis been performed appropriately and rigorously? 

Reviewer #1: Yes

4. Have the authors made all data underlying the findings in their manuscript fully available?

Reviewer #1: Yes

5. Is the manuscript presented in an intelligible fashion and written in standard English?

Reviewer #1: Yes

6. Review Comments to the Author

Reviewer #1: The authors have consistently improved their manuscript. All questions I addressed have been answered and/or taken into account in the revised manuscript. However, still one important aspect remains unaddressed in the revised MS.

Original question from the reviewer. L214-216. Why the authors pooled triplicates samples of DNA extraction in sediments? Keeping replicates separately could have helped to correct the imbalance between sites having very poor scores (n = 5) and good scores (n = 37).

Answer from the authors. We pooled before sequencing, and the purpose was to get as broad a representation as possible from a reach before sequencing. Our sampling unit was the stream reach, and therefore each replicate was not a replicate, but a subsample. There was no loss of power.

I understand the point of view of the authors. However, their choice of pooling PCR amplicons needs to be justified in the manuscript. The explanation you gave above would be pertinent according to your dataset “Doing otherwise would have been a large scale mismatch because the MBSS benthic macroinvertebrate samples are also pooled from several subsamples throughout the same reach”.

But further, the overall message of the MS is that monitoring of microbial communities in sediments can be considered as a complementary tool to assess the ecological condition of streams. Then, you need to precise how sediment samples were exaclty taken (in terms of granulometry, centre/edge of stream cannel...among others) to guide other researchers when applying your strategy. As you certainly know bacterial functions and diversity in stream sediments can change drastically depending of the conditions cited above. These aspects needs to be clearly defined in the M&M section of your MS.

7. PLOS authors have the option to publish the peer review history of their article (what does this mean?). If published, this will include your full peer review and any attached files.

Reviewer #1: No

---

## [Author Response · Author response to Decision Letter 1]

16 Jul 2020

please see the attached response to reviewers document

---

## [Editor Report · Decision Letter 2]

17 Jul 2020

Microbial Communities Can Predict the Ecological Condition of Headwater Streams

PONE-D-19-23872R2

Dear Dr. Hilderbrand,

We’re pleased to inform you that your manuscript has been judged scientifically suitable for publication and will be formally accepted for publication once it meets all outstanding technical requirements.

Kind regards,

Jean-François Humbert

Academic Editor

PLOS ONE
---

## [Editor Report · Acceptance letter]

23 Jul 2020

PONE-D-19-23872R2 

Microbial Communities Can Predict the Ecological Condition of Headwater Streams 

Dear Dr. Hilderbrand:

I'm pleased to inform you that your manuscript has been deemed suitable for publication in PLOS ONE. Congratulations! Your manuscript is now with our production department. 

Kind regards, 

on behalf of

Dr Jean-François Humbert 

Academic Editor

PLOS ONE